# JAK-inhibitors and risk on serious viral infection, venous thromboembolism and cardiac events in patients with rheumatoid arthritis: A protocol for a prevalent new-user cohort study using the Danish nationwide DANBIO register

Maria Luisa Faquetti [1☯], Enriqueta Vallejo-Yagüe [1,2☯], René Cordtz[3,4], Lene Dreyer [3,4,5‡], Andrea M. Burden [1,6‡]*

1 Institute of Pharmaceutical Sciences, ETH Zurich, Zurich, Switzerland, 2 Institute of Primary Health Care (BIHAM), University of Bern, Bern, Switzerland, 3 Center of Rheumatic Research Aalborg, (CERRA), Aalborg University Hospital, Aalborg, Denmark, 4 DANBIO Register, Aalborg, Denmark, 5 Clinical Institute, Aalborg University, Aalborg, Denmark, 6 Leslie Dan Faculty of Pharmacy, University of Toronto, Toronto, Canada

☯ These authors contributed equally to this work.
‡ LD and AMB also contributed equally to this work.
* andrea.burden@pharma.ethz.ch

## Abstract

Janus Kinase inhibitors (JAKis) are targeted synthetic disease-modifying antirheumatic drugs and represent an important alternative to treat patients with moderate to high rheumatoid arthritis (RA) disease activity. Safety concerns associated with increased risk for venous thrombo-embolism (VTE), serious viral infection, and, more recently, major adverse cardiovascular events (MACE) in JAKi users have emerged worldwide. However, as the exact mechanisms to explain these safety concerns remain unclear, the increased risk of VTE, MACE, and serious viral infection in JAKi users is heavily debated. In light of the need to enrich the safety profile of JAKis in real-world data, we aim to quantify the incidence and risk of MACE, VTE, and serious viral infections in RA patients registered in the Danish DANBIO registry, a nationwide registry of biological therapies used in rheumatology. Therefore, we will conduct a population-based cohort study using a prevalent new-user design. We will identify all RA patients in the DANBIO, ≥ 18 years old, receiving a JAKi or a tumor necrosis factor α inhibitor (TNF-αi) from January 2017 to December 2022. Prevalent and new users of JAKis will be matched to TNF-αi comparators with similar exposure history using time-conditional propensity scores (TCPS). We will describe the cumulative incidence of the outcomes (VTE, MACE, serious viral infection) in each exposure group (JAKi users; TNF-αi users), stratified by outcome type. Additionally, the Aalen-Johansen method will be used to estimate the time-to-event survival function stratified by outcome type. We will also estimate the hazard ratio (HR) with 95% confidence interval (CI) of each outcome in both exposure groups using the time-dependent Cox proportional hazards model. Results will enrich the safety profile of JAKis in real-world data.

**Data Availability Statement:** Due to restrictions imposed by the data provider, Statistics Denmark, the authors are not permitted to share the deidentified data. Data requests for future use will be handled by Statistics Denmark, (https://www. dst.dk/en). Additionally, further information on how to access the DANBIO data is available on their website (https://danbio-online.dk/front-page).

**Funding:** The author(s) received no specific funding for this work.

**Competing interests:** RC has received a consultant fee from Galapagos and is employed by IQVIA outside of the present work. LD has received research grant (paid to her institution) from BMS outside the current manuscript. LD is a member of the steering committee of the Danish Rheumatology Quality Registry (DANBIO, DRQ), which receives public funding from the hospital owners and funding from pharmaceutical companies. MLF, EVY, AMB declare no competing interests. This does not alter our adherence to PLOS ONE policies on sharing data and materials.

## Introduction

Janus Kinase inhibitor (JAKi) drugs are novel targeted synthetic disease-modifying antirheumatic drugs (tsDMARDs) and represent an important alternative to treat patients with moderate-to-serious rheumatoid arthritis (RA) disease [1]. The JAKis target kinases of the JAK family (JAK1, JAK2, JAK3, and non-receptor tyrosine-protein kinase TYK2), inhibiting the production of multiple pro-inflammatory cytokines, such as interleukin (IL)-6, IL-10, and interferon IFN-γ [2, 3]. For the treatment of RA, JAKi drugs with different affinities for one or more JAK enzymes have been approved worldwide, such as tofacitinib, baricitinib, and upadacitinib. While tofacitinib, for example, preferentially inhibits JAK1 and JAK3, baricitinib is designed to target JAK1 and JAK2, and upadacitinib is selective for JAK1 [3–5]. However, safety concerns on the increased risk of serious viral infections, venous thromboembolism (VTE), and major adverse cardiovascular events (MACE) associated with JAKis have emerged worldwide, regardless of their selectivity [6–12].

Patients with rheumatoid arthritis (RA) have increased risk of serious infection, thrombosis and cardiovascular risk factors [13–15]. An increasing amount of literature suggests that the use of JAKi drug may further increase the risk of cardiovascular diseases (e.g., increased serum lipid levels) and, thus, potentially increased the risk of MACE [11, 16–18]. Nevertheless, the underlying mechanism of cardiovascular outcomes due to JAKis use is not fully understood. Recent analyses of thromboembolic events as suspected adverse drug reactions for JAKis using real-world data support the need to investigate further this potential safety issue on different drugs of this class [8, 19]. While, conversely, integrated analyses of randomized clinical trials (RCTs) of JAKis did not reveal elevated risk of VTE [10, 12, 20, 21], safety assessment in RCTs is limited by the study conditions (e.g., time restrictions, population inclusion and exclusion criteria) and, therefore, it may not represent a real-world clinical setting. Thus, due to the paucity of evidence on the safety of JAKis at a population level, safety concerns regarding VTE and MACE associated with JAKis remain.

Therapies targeting the JAK family of enzymes may interfere with normal antiviral response, including inhibition of IFN-γ activity and may potentially further increase the risk of infection and reactivation of viral infectious diseases in RA patients. Pooling data from tofacitinib RCTs revealed an increased incidence rate of herpes zoster (HZ) than the observed rate in patients on biological disease-modifying antirheumatic drugs (bDMARDs) [10]. Similarly, Curtis et al. reported a 2-fold increased risk for HZ in RA patients treated with tofacitinib compared to RA patients using biologics (including, but not exclusively, tumor necrosis factor α inhibitor [TNF-αi] drugs) in a real-world study [22]. Nevertheless, there is a lack of real-world evidence on the risk of other serious infections (e.g., cytomegalovirus and Epstein-Barr virus) in patients with RA using JAKi drugs.

In light of the need to enrich the safety profile of JAKis in real-world data, we aim to quantify the incidence and risk of MACE, VTE and serious viral infections in RA patients registered in the Danish rheumatologic database (DANBIO).

## Materials and methods

### Study objectives

The study objectives are to investigate the incidence (frequency) and risk of (1) major adverse cardiovascular events (MACE), (2) venous thromboembolism (VTE), and (3) serious viral infection (or reactivation) associated to the use of JAKis compared to TNF-αis among RA patients.

## Study design and setting

We will conduct a population-based cohort study using a prevalent new-user design [23]. Clinical data, medications, and outcomes will be obtained for patients enrolled in the Danish Rheumatologic database (DANBIO) between January 1st, 2017, and December 31st, 2022 (or the latest available in the database). Since 2006, DANBIO registry collects data prospectively using a web-based system used in routine care at Danish hospital Departments of Rheumatology or private rheumatologic clinics.

## Data source

The study will use data from the DANBIO with additional information obtained from the Danish National Patient Registry, the Danish Civil Registration System in Denmark, Danish National Database of Reimbursed Prescriptions, and the Danish National Prescription Registry. The unique civil registration number (CPR number) from the Danish Civil Registration System [24] enables linkage between these national data sources on an individual-based level.

The DANBIO registry provides information on RA diagnosis, patient demographics (sex and age), patient characteristics (e.g., smoking status), RA disease duration, RA disease activity score [e.g., 28-joint disease activity score (DAS28)], health assessment questionnaire (HAQ), biologic markers [e.g., rheumatoid factor (RF) and cyclic citrullinated peptides] and inflammatory markers [e.g., C-reactive protein (CRP)], patient-reported outcomes [e.g., visual analogue scale (VAS) on pain], other clinical endpoints, and anti-rheumatic medication (with start and stop dates) including conventional synthetic disease-modifying antirheumatic drugs (csDMARDs), corticosteroids, tsDMARD, and biologics (e.g., TNF-αi and non- TNF-αi biologics). No distinction between biologics and biosimilars will be done in this study.

The Danish National Patient Registry provides information on disease diagnoses [classified by the International Classification of Diseases, 10th revision (ICD-10) codes] [25]. Moreover, it has complete data regarding hospitalizations and outpatient care. Thus, information on comorbidities and outcomes (e.g., cardiovascular diseases, VTE, and serious viral infections) will be collected from the Danish National Patient Registry. Additionally, the Danish National Prescription Registry provides data on non-rheumatic medication [identified by Anatomical Therapeutic Chemical Classification (ATC) codes] [26]. Finally, the Danish Civil Registration System provides information on vital status, sex, date of birth, and migration date and status.

The Danish healthcare model is similar to other countries such as Canada, Italy, or France, where there is a single-payer Universal healthcare system. Thus, patients in Denmark receive bDMARDs or JAKi at no cost as they are provided free of charge within the hospital setting. Regarding access to medication, JAKis are eligible as a third-line therapy option, after csDMARDs and bDMARDs. Thus, patients receiving JAKis within Denmark will have failed or be contraindicated for bDMARDs prior to receiving a JAKi. Finally, clinical practice in Denmark traditionally follows the European Alliance of Associations for Rheumatology (EULAR) guidelines [27], and therefore, we expect our results to be representative to other countries following EULAR guidance.

## Ethics statement

The study complies with the Declaration of Helsinki. Registry-based research does not require approval from the Danish National Research Ethics Committee. The study was approved by the DANBIO (ref. DANBIO-2018-11-30) steering committee and registered at the North Region's inventory (ref. P-2018–182).

### Patient consent for publication

DANBIO is approved as a national quality registry and by the Danish Data Protection Agency, which is based on the Act on Processing of Personal Data and ensures that data security, protection and individual rights, are dealt with correctly. This constitutes the necessary legal requirements, and informed consent is not required per Danish legislation [28]. Therefore, informed patient consent is not needed to register or participate in the DANBIO registry [29].

### Data management plan

To protect the participants' privacy and to maintain confidentiality, only completely anonymized data from patients registered in DANBIO will be included in the study. Thus, the authors will have no access to information that could identify participants during data analysis. The data will be analyzed securely with remote access to Statistics Denmark. Also, only aggregated data with sufficient masking (i.e., $\geq 3$ events per cell) will be made available on reasonable request.

### Study population

**Base cohort.** We will identify all RA patients in DANBIO database, $\geq 18$ years old, receiving a JAKi or a TNF-αi (i.e., incident and prevalent users) between January 1st, 2017 and December 31st, 2022 (or latest available in the database).

TNF-αi users with previous use of a JAKi, patients missing age or sex, and patients with a diagnosis of cancer (except non-melanoma skin cancer) ever before $T_0$ (i.e., study cohort entry date for JAKi users and the corresponding matching date for TNF-αi users) will be excluded. Two cohorts will be created to evaluate the outcomes of interest ([1] MACE or VTE, and [2] serious viral infections) independently. Thus, additionally, for the population that will lead to MACE or VTE base cohort, patients with a diagnosis of mitral stenosis, valvular heart diseases, valve replacement, heart transplantation, MACE- or VTE-events (i.e., defined as per the outcomes) ever before $T_0$ will not be eligible for matching in the exposure set. For the serious viral infection base cohort, a diagnosis for serious viral infection (i.e., defined as per the outcomes), or antiviral therapy (acyclovir, valaciclovir and famciclovir) within the 365 days prior $T_0$ (i.e. wash-out period) will not be eligible for matching in the exposure set. Patients with a diagnosis of human immunodeficiency virus (HIV) ever before $T_0$ in the serious viral infection cohort will be excluded.

A graphical representation of base cohorts for MACE/VTE outcomes and Serious viral infection, depicting inclusion and exclusion criteria, is shown in **Figs 1** and **2**, respectively. Starting from patients in the DANBIO registry database, we may define subsequent key population steps in this study: base cohorts, exposure sets, and study cohorts. These are explained in detail later in the protocol, here in brief: As illustrated in **Figs 1** and **2**, a base cohort will be obtained after applying the inclusion and exclusion criteria. This will include JAKis and TNF-αi initiators (i.e., potential controls) starting the drug treatment within the study period. From the base cohort, JAKis users will be identified as members of the JAKi group and subsequently matched with selected TNF-αi users within their exposure set. For that, the exposure set for each JAKi user will include a pool of TNF-αi users from the base cohort who are eligible to be their controls (i.e., due to similar exposure until the time of JAKi start). Notably, the exposure sets are the distinction between prevalent and incident users of JAKs. Lastly, the matched JAKi users and TNF-αi users with similar treatment history at the JAKi start constitute the study cohorts.

A complete list for RA diagnoses (i.e., inclusion criteria) using the International Classification of Diseases 10th revision (ICD-10) codes is available in **S1 Table**. A Complete list of ICD-

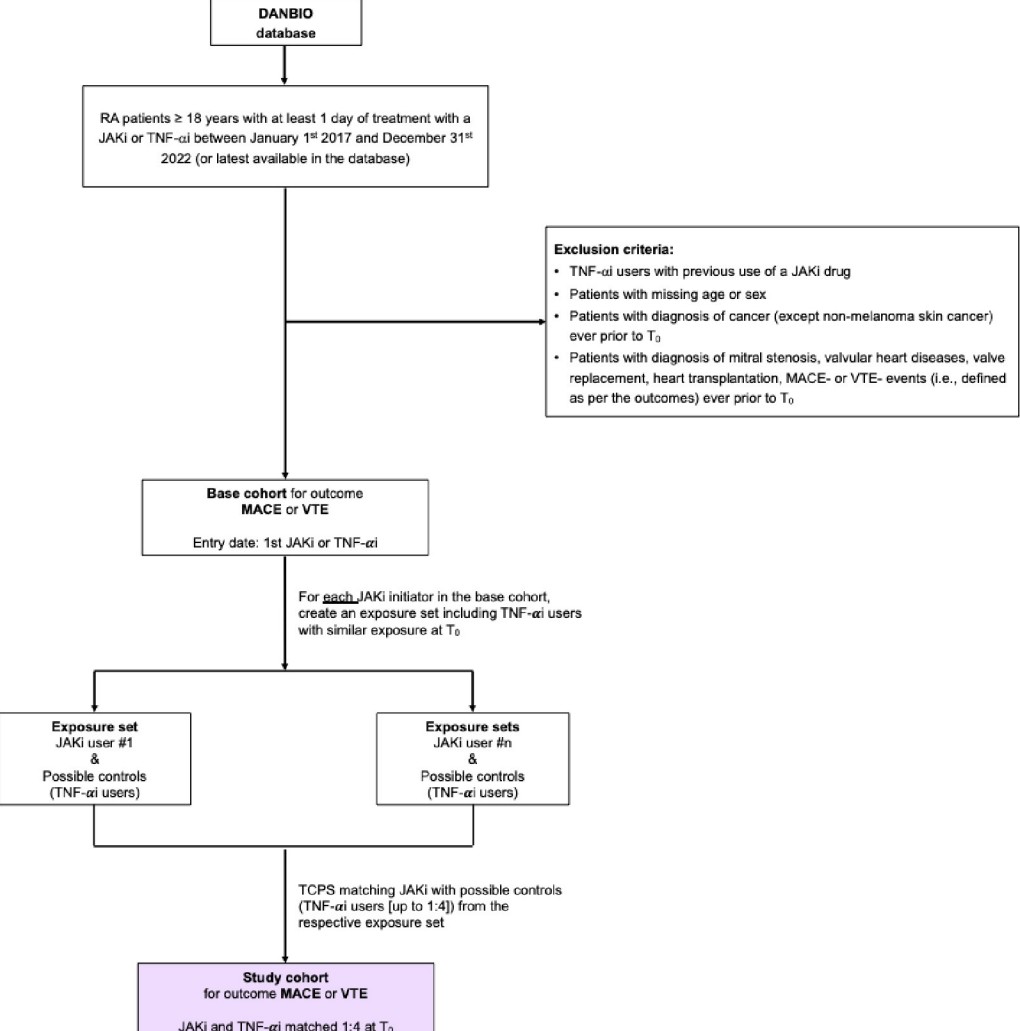

**Fig 1. Base cohorts for MACE and VTE outcomes, inclusion and exclusion criteria, exposure sets, and study cohorts.** Abbreviations: RA Rheumatoid Arthritis; JAKi Janus Kinase inhibitor; TNF-αi Tumor Necrosis Factor α inhibitor; DANBIO Danish Rheumatologic database; MACE major adverse cardiovascular events; VTE venous thromboembolism. TCPS time-conditional propensity score; $T_0$ will be defined as the study cohort entry date for JAKi users (i.e., first start of JAKi treatment) and the corresponding matching date for TNF-αi users.

10 codes for diagnosis and the Anatomical Therapeutic Chemical (ATC) classification codes for antiviral therapy used as exclusion criteria is available in **S2 Table**.

The base cohorts to evaluate the outcomes of interest will be created including patients with at least one day of treatment with JAKi or TNF-αi during the study period. The base cohort entry date for the two cohorts will be defined as the date of treatment initiation (i.e., 'treatment start date') for a JAKi or a TNF-αi. Patients will be allowed to enter the base cohort a maximum of two times, first with a TNF-αi and second with a JAKi (but not vice versa given the use of the prevalent new-user design). Use of JAKis prior to the study period is not expected due to later approval of these drugs in Europe. Use of TNF-αi or other bDMARD prior to the study period will not be penalized.

The base cohort will therefore include:

1. Incident new-users of JAKis (i.e., new user of JAKi without prior use of TNF-αi);

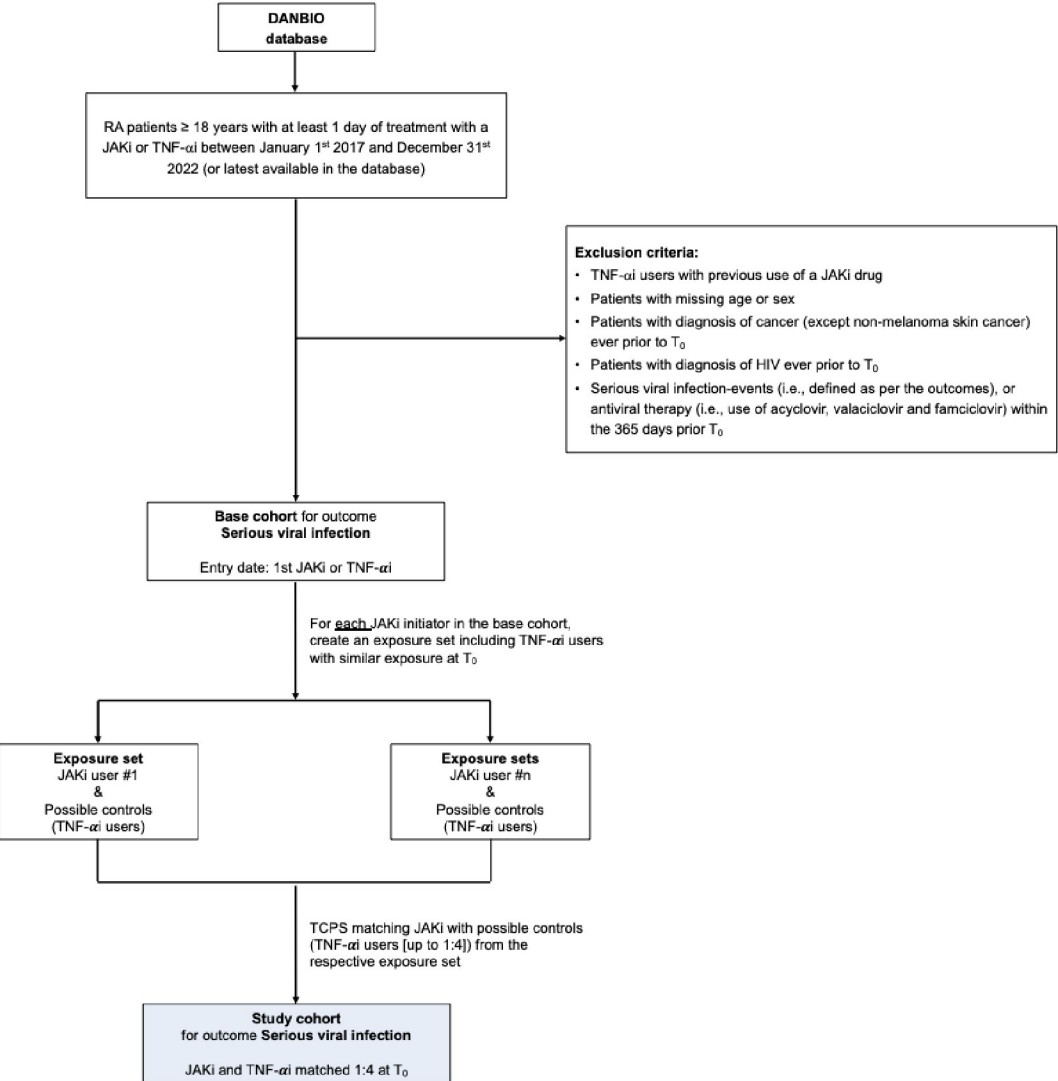

**Fig 2. Base cohorts for the outcome of serious viral infection, inclusion and exclusion criteria, exposure sets, and study cohorts.** Abbreviations: RA Rheumatoid Arthritis; JAKi Janus Kinase inhibitor; TNF-αi Tumor Necrosis Factor α inhibitor; HIV Human Immunodeficiency Virus; DANBIO Danish Rheumatologic database. TCPS time-conditional propensity score; $T_0$ will be defined as the study cohort entry date for JAKi users (i.e., first start of JAKi treatment) and the corresponding matching date for TNF-αi users.

2. Prevalent new-users of JAKis, also called switchers (i.e., new user of JAKis with previous use of TNF-αi drug);

3. Incident new-users of TNF-αi (i.e., new user of TNF-αi without prior use of TNF-αi);

4. Prevalent new-users of TNF-αi (i.e., new user of TNF-αi with prior use of TNF-αi).

Subsequently, exposure sets will be created for each JAKi user as described in the below subsections, and as depicted in **Fig 3**.

**Exposure sets.** Among the patients included in the base cohorts, a set of potential controls (i.e., exposure set) will be defined for each JAKi user at the time of the JAKi treatment initiation ($T_0$). The exposure set (i.e., potential controls) for each JAKi user will include TNF-αi users with similar exposure history at $T_0$, who will be chosen using time-based exposure sets.

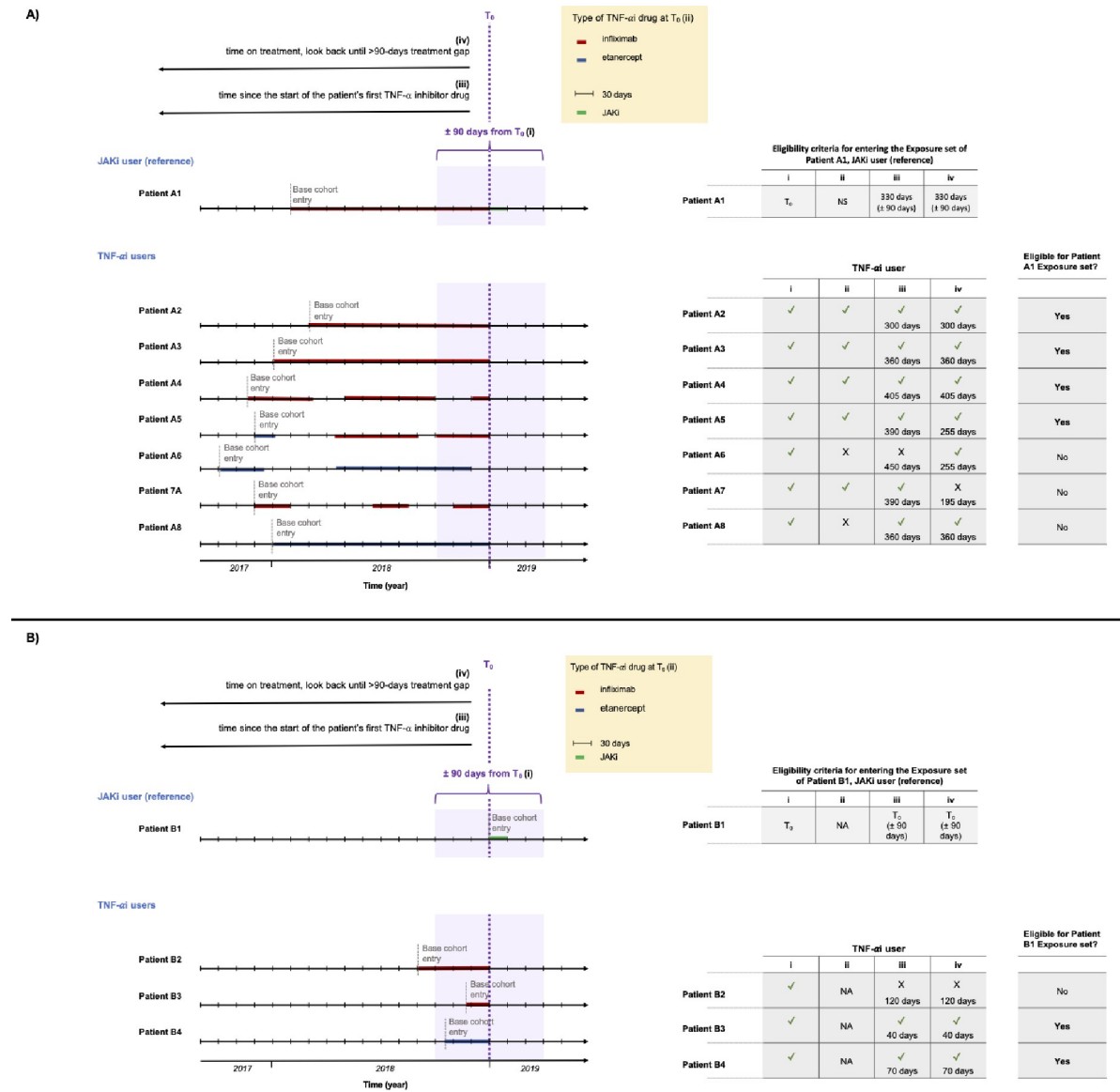

**Fig 3. Graphical description of the time-dependent exposure set for Janus Kinase inhibitors (JAKis) users.** (A) We depict an example of potential controls (patients A2, A3, A4, A5) to be included in the exposure set for a JAKi user with previous TNF-αi use (patient A1), and three examples (patients A6, A7, and A8) who are not eligible to be controls for that JAKi user based on: (i) calendar time (TNF-αi use within ± 90 days from $T_0$), (ii) type of TNF-αi at $T_0$ (not applicable for JAKis users with no previous use of TNF-αi), (iii) time since the start of the patient's first TNF-αi drug within a difference of ± 90 calendar days (i.e., look back period from study cohort entry date up to base cohort entry), and (iv) duration (days) of the most recent TNF-αi treatment course, starting looking from $T_0$ and continue backwards in time until a period free of treatment > 90 days. (B) Potential controls (patients B3 and B4) to be included in the exposure set for a JAKi user with no previous TNF-αi use (patient B1), and one example (patient B2) who is not eligible to be a control for that JAKi user based on the requirements above (i), (iii), and (iv). While infliximab and etanercept were used to exemplify patients using different TNF-α drugs, the list of TNF-αis used in the study is not restricted to these two drugs. Abbreviations: NA not applicable; JAKi Janus Kinase inhibitor; TNF-αi tumor necrosis factor α inhibitor; $T_0$ will be defined as the study cohort entry date for JAK inhibitor users and the corresponding matching date for TNF-αi users.

Exposure sets will be defined by i) calendar time (TNF-αi drug use within ± 90 days from $T_0$), (ii) type of TNF-αi drug, (iii) time since the start of the first TNF-αi drug within a difference of ± 90 calendar days (i.e., look back period from study cohort entry date up to base cohort entry), and (iv) duration (days) of the most recent TNF-αi treatment course, starting looking from $T_0$ and continue backwards in time until a period free of treatment > 90 days to

account for direct and delayed switchers. A graphical description of the time-dependent exposure set for an example of JAKi prevalent new-user (patient A1) is depicted in **Fig 3A**. The figure shows several TNF-αi users from the base cohort as potential controls (patients A2 to A8). Additionally, in **Fig 3B**, potential controls to be included in the exposure set for a JAKi user with no previous TNF-αi use (patient B1). TNF-αi users may enter more than one exposure set, at different times, eventually matching the respective JAKi users $T_0$. Note that the base cohort entry may be previous to the study period for TNF-αi users.

**Study cohort.**    Within the corresponding exposure set, each JAKi user will be matched to one TNF-αi user at $T_0$ using time-conditional propensity scores (TCPS). Therefore, the study cohorts will include the matched JAKis—TNF-αi pairs, as depicted in **Figs 1** and **2**. $T_0$ will be defined as the study cohort entry date for JAKi users, and the corresponding matching date for TNF-αi users. Note that for JAKi incident new-users, the base cohort entry date and $T_0$ day will be identical. Patients will be allowed to enter the study cohort a maximum of two times, first with a TNF-αi and second with a JAKi (but not vice versa given the use of the prevalent new-user design).

**Sample size considerations.**    The number of outcome events required in each group (JAKi and TNF-αi) was estimated by outcome type using current literature. All calculations assume 80% power and alpha 0.05.

For estimating the risk of infection, Curtis and collaborators previously found an adjusted HR for HZ with tofacitinib compared to biologic agents (including TNF-αi and non- TNF-αi) of 2.01 (95% CI 1.40 to 2.88) [22]. Assuming (i) an equivalent risk of HZ for the composite outcome of serious viral infections used in this study, and (ii) that the incidence risk is similar for baricitinib and upadacitnib to tofacitinib, 64 events are required [30].

Molander and collaborators previously assessed incident VTE events with JAKis compared TNF-αis in patients with RA [31]. In their study, the adjusted HR for VTE was 1.73 (95% CI 1.24 to 2.42) for JAKi users compared to TNF-αi users. Therefore, 104 events are required [30]. The adjusted HR for MACE in tofacitinib users compared to TNF-αi users found by Ytterberg and collaborators in RA patients aged >50 year and having at least one cardiovascular risk factor was 1.33 (95% CI 0.91 to 1.94) [11]. Thus, assuming a similar risk of MACE for baricitinib and upadacitnib to tofacitinib, 386 events are required given the suggested relative hazard [30].

In the event that study is not adequately powered, we will also report the unadjusted incident rates and incidence rate ratio's as the topic is of high clinical and regulatory importance. Additionally, since this is a real-world study in registry data, every patient in DANBIO database fulfilling the eligibility criteria will be included in the study.

**Study exposure.**    Patients will be classified into one of the two mutually exclusive groups at entry into the study cohort: current use of JAKis or current use of TNF-αis. Exposure will be defined using the as-treated approach (i.e. defined at study cohort entry and considered time-fixed). A complete list of ATC codes for JAK and TNF-αi is available in the **S3 Table**.

**Follow-up and censoring.**    We will follow all patients in the study cohort from the study cohort entry date until the earliest of the following: occurrence of first ICD-10 code for an outcome event, end of study period, or censoring as a result of change in the exposure status. Change in exposure status will be defined as (i) treatment discontinuation (defined by the earliest of the following: a recorded treatment discontinuation date, or at the beginning of a treatment gap of >90 days) or (ii) switching from a JAKi to a non-JAKi drug, or (iii) switching from a TNF-αi to a non-TNF-αi drug, or (iv) death. We will accept a permissible gap of up to 90 days between courses of the same exposure treatment. During follow-up, the TNF-αis will be treated as a class.

**Study outcome(s).** MACE, VTE and serious viral infections will be identified using the ICD-10 codes in the Danish National Patient Registry. MACE will be defined as a composite of myocardial infarction, stroke, and cardiovascular (CV) death. VTE will be defined as a composite of pulmonary embolism, deep vein thrombosis, and other embolisms. Serious viral infection (or viral reactivation) will be defined as a composite of herpes zoster (HZ), cytomegalovirus (CMV), and Epstein-Barr. A complete list of outcome codes is available in the **S4 Table.**

**Covariates.** A list of covariates to be assessed at $T_0$ is provided in **S5 Table**. Comorbidities will be identified in the Danish National Patient Registry as diagnosed in the previous 10 years from the study cohort entry date, or otherwise specified in **S5 Table**, in the 'comments' column. Indicators of disease severity (e.g., disease activity score-28 [DAS28] and the Stanford Health Assessment Questionnaire-Disability Index [HAQ-DI]) and disease duration (i.e., time since RA diagnosis) will be collected at at $T_0$, or will be defined as the closest value looking back up to 90 days prior to $T_0$. Comedication will be defined as drug use at $T_0$, looking back up to 180 days prior to $T_0$ or otherwise specified in **S6 Table**. Previous immunization for HZ will be assessed at $T_0$, looking back up to 5 years. Ascertainment periods for covariates in S5 Table and S6 Table were chosen based on clinical and methodological rationale.

Note that TNF-αi users may be included in none, one, or more exposure sets. In the case of a TNF-αi user included in several exposure sets, covariates for these patients will be assessed in each exposure set, and therefore a TNF-αi user may have different covariate values in different exposure sets. The complete lists with ATC codes for comedication and HZ immunization are available in the **S6** and **S7 Tables**, respectively.

**Handling missing data.** The number or frequency of missing data in relevant variables will be indicated for transparency purposes. For matching purposes, missing values on disease duration, HAQ-DI, DAS28, RF factor, anti-cyclic citrullinated peptides (CCP) status, smoking status, and alcohol consumption, will be considered as a separate category. Nevertheless, we do not expect an elevated frequency of missing data for key variables due to the high completeness of data in DANBIO database [29, 32]. For comorbidities (e.g., depression, diabetes, and psoriasis) we will assume that the absence of an ICD-10 code means that the patient was not diagnosed with that condition. For comedication we will assume that if the ATC code is not recorded, the patient does not use the drug.

**Time-conditional propensity score (TCPS) matching.** We will construct TCPS separately for incident and prevalent new-users by using conditional logistic regression with time-varying covariates, including the covariates provided in **S5 Table**, and stratifying by exposure set (therefore "conditional" on exposure set). Hence, an individual may have different scores for different exposure sets they enter, depending on the time of entry.

We will match JAKi users to TNF-αi users within exposure sets on nearest TCPS and on a variable ratio (i.e., one JAKi user will be matched with up to four TNF-αi controls), without replacement and in chronological order. However, if more than 10% exposure sets have no suitable match available after trimming the distribution of TCPS, we will perform matching with replacement. Note that a patient may be matched as TNF-αi user to a JAKi user and later during the follow-up initiate with a JAKi. In this case, the subject will be included as a new user of a JAKi from the point of switch onwards and a matched comparator will be identified at the switching point. While the TCPS for incident new users will be the probability of initiating a JAKi compared to a TNF-αi, for prevalent new-users it will be the probability of a patient switching treatment from TNF-αi to JAKi.

A Caliper of 0.06 standard deviation (SD) of logit TCPS within each exposure set will be used to define the range of estimated TCPS within which to select the TNF-αi patient. Unbalanced variables will be adjusted in the statistical analysis (Cox proportional hazard analysis).

To satisfy the positivity assumption, we will exclude exposure sets where the TCPS of patients treated with TNF-αi are not within the range of the TCPS distribution of the corresponding JAKi exposure set.

## Statistical analysis

**Primary analysis.** We will assess patients' characteristics and covariate distribution among the JAKi users and TNF-αi users included in exposure sets at $T_0$. This will be performed before and after TCPS-matching in the two exposure groups in each cohort. Similarly, the mean follow-up time in person-years before, to evaluate the quality of the TCPS matching and potential imbalanced censoring. The accuracy of matching among covariates will be assessed using the absolute value of the standardized differences (with a value of 0.1 or more to be considered important). Categorical variables will be presented as counts and percentage, and continuous variables as mean and SD or median and interquartile range (IQR).

The study analyses will be performed separately for each cohort ([1] MACE or VTE, and [2] serious viral infection) on an outcome basis. We will describe the cumulative incidence of outcome new events in each exposure group, stratified by outcome type (MACE, VTE or serious viral infection). Aalen-Johansen method will be applied to estimate the survival function of time-to-event from study cohort entry, stratified by outcome type (MACE, VTE or serious viral infection).

Subsequently, the Cox proportional hazard model will be used to conduct the statistical analysis between matched pairs of patients exposed to JAKis and patients exposed to TNF-αis in the two cohorts. Thus, we will estimate the hazard ratios (HR) with 95% confidence interval (CI) of each outcome (MACE, VTE and serious viral infection) associated to JAKis use compared to TNF-αis use. Crude incidence rates and crude and adjusted HR estimates and the corresponding 95% confidence intervals (95% CIs) will be presented for each outcome MACE, VTE and serious viral infection. The Cox proportional hazard analysis will be performed using the maximum partial likelihood method for estimation with a Newey-West covariance matrix to account for the same patient contributing TNF-αi-exposed person time and JAKi-exposed person time and if matching with replacement is used.

**Subgroup analysis.** Secondarily, we will perform a subgroup analysis by sex, similarly to above described for the overall study cohorts. However, if the reduced sample size does no longer enable to performed these subgroup analyses as planned, we will alternatively only report the incidence risk ratio (crude and adjusted).

Other sub-group analyses would only be performed if sample size allows it (e.g., JAKi drug, dose, age).

**Sensitivity analysis.** We will repeat our primary analysis (i) stratified by follow-up period ($\leq 1$ year and $> 1$ year) to examine the impact of follow-up duration in our results, (ii) varying the treatment gap to define continuous drug use to 0 and 180 days, (iii) varying the period free of treatment in the previous 365 days varying from 0 and $\leq 180$ days to estimate the cumulative time on TNF-α treatment (in days) at $T_0$.

**Other analysis for evaluating the impact on unmeasured confounding.** We will apply the E-value as initial method to assess severity of unmeasured confounding (no unmeasured confounding assumption by the TCPS) [33]. A strength of the E-value approach is that it makes minimal assumptions regarding the structure of unmeasured confounding (e.g., it does not assume the unmeasured confounder is binary), nor does it assume that there is a single unmeasured confounder). It also does not require assumptions regarding the prevalence or distribution of the unmeasured confounder(s).

A large E-value in context of study design and existing TCPS method for confounding control, indicates that, besides reporting E-value, further sensitivity analysis may not be needed.

**Table 1. Timeline of the study.**

| Year | Activity |
| --- | --- |
| 2023 | Acquiring the DANBIO data |
| 2023 | Data Analysis |
| 2024 | Publication of results |
| 2024 | Conference Presentations |

Conversely, small E-value can be due to an unmeasured confounder that is (i) a single binary variable which the prevalence is quite high or quite low, or (ii) not a single binary variable or prevalence of a single binary confounder is unknown. If the first occurs, we will perform the rule-out method [34] as second sensitivity analysis, and if the latter occurs (or moderate or large impact is observed from the rule-out analysis), we will further evaluate the impact of unmeasured confounders using, for example, propensity score calibration [35].

**Dissemination plan.** Studies from this project will be submitted for publication in high-impact international journals and be presented at international rheumatology and pharmacoepidemiology conferences.

**Status and timeline of the study.** Data analysis will start when DANIO data is available. The data is expected to be available at any time from January 31st, 2023. The timeline of the study is depicted in **Table 1**.

## Discussion

This population-based cohort study aims to investigate the incidence and risk of MACE, VTE, and serious viral infection associated with the use of JAKis compared to TNF-αis among RA patients in real-world clinical practice. Moreover, to our knowledge, this will be the first study in DANBIO database to investigate the risk of MACE, VTE and serious viral infection (or reactivation) associated with JAKis in RA patients.

When it comes to JAKis approved to treat RA, information from safety trials have let to warnings and precautions on the labels of JAKi [36, 37]. For example, the post-marketing ORAL Surveillance study, which compared tofacitinib versus TNF-αis in RA patients aged 50 years and older with cardiovascular risk factors, have led the recommendation for using JAKis with caution [11, 37]. Although RCTs are the gold standard to determine drug safety and provide initial data on safety risks, it is limited by the study conditions (e.g., time restrictions, population inclusion and exclusion criteria). Therefore, RCTs may miss the various nuances of different patients in a real-world clinical setting. For example, patients included in clinical trials often have less comorbidities and receive less polypharmacy compared to the overall population, due, potentially presenting lower risk of infection and fewer cardiovascular risk factors, resulting in lower adverse event rates. Thus, real-world evidence on the matter is of interest as it provides additional insights into the real-world incidence and patterns of adverse events.

Nevertheless, pharmacoepidemiologic studies on real-world data to assess safety of JAKis are challenging, due to the common prescription of JAKi after previous failure of bDMARD (e.g., TNF-αi). While a traditional new-user design if often preferable to investigate safety concerns (because it mimics RCT design in real-world setting), this applied to our research question would lead to the exclusion of the majority of JAKi users (i.e., exclusion of JAKi users with previous use of TNF-αi). This would result in strong reduction of sample size and, importantly, reduced generalizability of study results. Thus, a key strength of this study is the methodological approach to overcome this challenge, by using the prevalent new-user design [23].

At the same time the prevalent new-user design enables the inclusion of most JAKi users, it reduces potential confounding by indication and the potential consequences of depletion of susceptible patients (i.e., prevalent-user bias) by time-conditional propensity score matching JAKi users with TNF-αi controls on previous history.

Another strength of this study is the use of DANBIO database. The register has high Danish nationwide coverage of patients with rheumatic diseases and high completeness on key data variables, such as RA diagnosis, disease scores, biomarkers, and history of cs/bDMARD therapy. Moreover, DANBIO database can be linked to other data sources in Denmark (e.g., the Danish National Patient Registry, a nationwide administrative registry that covers all hospitalizations and outpatient visits). Therefore, by using DANBIO database, this study will integrate the entire patient history to obtain a complete view of the patient history. This protocol has been designed for the DANBIO registry, and thus, a Danish patient population. Denmark has a universal healthcare system and follows the EULAR guidelines for treatment recommendations, and as such, it may have some inherent differences from other countries. For example, we may expect fewer socio-economic differences between our patient groups and will provide comparisons between our cohort and other published studies. However, one of the strengths of the study design is that the prevalent-new user design with propensity score matching addresses circumstances in which the study and reference drugs are not prescribed under the same criteria. While we would not expect substantially different results in different patient settings, it is possible that finding optimal bDMARD matches for JAK inhibitor patients may be more challenging in some populations.

Finally, despite the overall completeness and validity of outcome diagnoses in the Danish National Patient Registry [12], the potential for bias due to misclassification of outcomes should not be completely ruled out. This is an intrinsic limitation of RWD-based studies, as data is often collected for non-research purposes

## Supporting information

**S1 Table. List of diagnosis code using the International Classification of Diseases 10th revision (ICD-10) for inclusion criteria.**
(DOCX)

**S2 Table. List of diagnosis code using the International Classification of Diseases 10th revision (ICD-10) and drug therapy using the Anatomical Therapeutic Chemical (ATC) classification codes for antiviral therapy used as exclusion criteria.**
(DOCX)

**S3 Table. List of drug therapy using the Anatomical Therapeutic Chemical (ATC) classification codes for exposure definition.**
(DOCX)

**S4 Table. List of conditions using the International Classification of Diseases 10th revision (ICD-10) for outcome definition.**
(DOCX)

**S5 Table. List of covariates for Time-Conditional Propensity Score (TCPS).**
(DOCX)

**S6 Table. List of Anatomical Therapeutic Chemical (ATC) classification codes for definition of comedication.**
(DOCX)

**S7 Table. List of Anatomical Therapeutic Chemical (ATC) classification codes for definition of Herpes Zoster (HZ) immunization.**
(DOCX)

## Acknowledgments

We acknowledge the substantial contribution of the academic hospitals and private clinics and their physicians that report data to DANBIO.

## Author Contributions

**Conceptualization:** Andrea M. Burden.

**Funding acquisition:** Lene Dreyer, Andrea M. Burden.

**Methodology:** Maria Luisa Faquetti, Enriqueta Vallejo-Yagüe.

**Resources:** Lene Dreyer, Andrea M. Burden.

**Software:** René Cordtz.

**Writing – original draft:** Maria Luisa Faquetti, Enriqueta Vallejo-Yagüe.

**Writing – review & editing:** Maria Luisa Faquetti, Enriqueta Vallejo-Yagüe, René Cordtz, Lene Dreyer, Andrea M. Burden.

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
