## [Decision Letter · Decision Letter 0]

5 Jun 2023

PONE-D-23-07128JAK-inhibitors and risk on serious viral infection, venous thromboembolism and cardiac events in patients with Rheumatoid Arthritis: A protocol for a prevalent new-user cohort study using the nationwide DANBIO register.PLOS ONE

Dear Dr. Burden,

Thank you for submitting your manuscript to PLOS ONE. After careful consideration, we feel that it has merit but does not fully meet PLOS ONE’s publication criteria as it currently stands. Therefore, we invite you to submit a revised version of the manuscript that addresses the points raised during the review process.

We look forward to receiving your revised manuscript.

Kind regards,

Ryu Watanabe, M.D., Ph.D.

Academic Editor

PLOS ONE

“I have read the journal's policy and the authors of this manuscript have the following competing interests: RC has received a consultant fee from Galapagos and is employed by IQVIA outside of the present work. LD has received research grant (paid to her institution) from BMS outside the current manuscript.  She is member of the steering committee of the Danish Rheumatology Quality Registry (DANBIO, DRQ), which receives public funding from the hospital owners and funding from pharmaceutical companies. MLF, EVY, AMB have declared that no competing interests exist.”

Reviewers' comments:

Reviewer's Responses to Questions

**Comments to the Author**

1. Does the manuscript provide a valid rationale for the proposed study, with clearly identified and justified research questions?

Reviewer #1: Yes

Reviewer #2: Yes

2. Is the protocol technically sound and planned in a manner that will lead to a meaningful outcome and allow testing the stated hypotheses?

Reviewer #1: Yes

Reviewer #2: Yes

3. Is the methodology feasible and described in sufficient detail to allow the work to be replicable?

Reviewer #1: No

Reviewer #2: Yes

4. Have the authors described where all data underlying the findings will be made available when the study is complete?

Reviewer #1: Yes

Reviewer #2: Yes

5. Is the manuscript presented in an intelligible fashion and written in standard English?

Reviewer #1: Yes

Reviewer #2: Yes

6. Review Comments to the Author

You may also provide optional suggestions and comments to authors that they might find helpful in planning their study.

Reviewer #1: The authors plan to study the prevalence, and incidence, of serious viral infections, VTE and cardiac events in the Danbio registry comparing JAKi with TNF treated patients.

Their methodology is quite sound and they have done a good job in presenting it.

However, and this is important, the reader is unable to judge the representativeness and generalizability of their approach. This is because the Danish health care system approves minimal disease criteria in order to be eligible to receive either a b or tsDMARD. These criteria differ across national health care systems. These European eligibility criteria are typically quite different from country to country and are very different from eligibility criteria in the US.

This is reflected by the very much different prevalence of prescribing both b and tsDMARDs in single national health care systems from country to country.

Therefore, the authors must present plans to supply data on: 1. Eligibility to receive both b and tsDMARDs in the Danish system; 2. Percentage of patients (prevalent and incident described separately) who are prescribed these agents with the overall percentage of patients treated in the health care system used as a denominator; and 3. Describe how the population studied in this selective population might affect the results they report.

That is, if only patients with more severe disease receive these agents in Denmark, while patients with much less (mean and median) standard disease metrics receive these same drugs in other countries then these differences are potentially very relevant and affect the generalizability and representativeness of other populations around the world who access these drugs with different baseline disease activity criteria. National health care systems are able to set their own minimal disease criteria based upon a policy of willingness to pay for different disease activity levels.

Reviewer #2: Thank you for giving me the opportunity to review this article. I have carefully read the article. Although there are important safety data from individual RTCs as well as from pooled analyses on RCTs, real-world data comparing safety of JAKi and TNFi are currently limited. Therefore, the aim of the study serves to provide valuable safety knowledge on JAKi treatment of RA patients.

The article deals with the study design and methodology of the planned prospective cohort study, and overall, the study design has been elaborately made to avoid confounding. However, some part of the description lacks detailed explanation.

I suggest that the followings be explained in more detail.

1. Different abbreviations (DNPR and NPR) were used for the same meaning in the “DATA SOURCE| section.

2. For Figure 1. incorporate the difference between MACE/VTE cohort and viral infection cohort.

3. Figure 1 should come after the paragraph “base cohort”.

4. In figure 2, only prevalent new user of JAKi and his or her potential controls are depicted. Please make it include the incident new users.

5. In figure 2, describing why patient 3 and 4 are not potential controls will help readers to understand the study cohort selectiotn process.

6. In “Base Cohort” section, the sentence “Note that the base cohort entry may be previous to the study period for TNFi users” will be better placed in “Exposure set” section since such discrepancy occurs when matching a TNFi user to a prevalent new user of a JAKi. Figure 2 will help understanding, which is also being placed in “Exposure set” section.

7. There are four levels of study populations: study population, base cohort, exposure sets, and study cohort, which makes the readers confusing. Please add explanations for the purpose of this multi-level populations “upfront” and psecify the role of each population.

8. Do authors distinguish switching between different TNFis? Authors did not specify whether they will censor at switching from one TNFi to a different TNFi, which makes me assume that the authors will treat TNFis as a class. However, in Figure 2, the exposure set condition includes types of TNFi.

9. Please present the PPV of each outcome definition for the DNPR. If there is no PPV made for ICD10 used with DNPR, describe possibility of the misclassification bias.

10. Follow-up and censoring

censoring as a result of change in the exposure status, which will be defined as (i) treatment discontinuation (defined by recorded treatment discontinuation or a gap of >90 days, in which case we assume stop at start/stop of the gap) please rephrase the underline part since it is hard to follow.

11. ascertainment periods for individual covariates are different. Briefly explain on which criteria each of the time period was set.

12. It is not clear whether the matching is fixed ratio (1:4) or variable ratio. Please specify.

7. PLOS authors have the option to publish the peer review history of their article (what does this mean?). If published, this will include your full peer review and any attached files.

Reviewer #1: No

Reviewer #2: No

---

## [Author Response · Author response to Decision Letter 0]

3 Jul 2023

please see attached response to reviewer document for a detailed response to the individual reviewer comments.

---

## [Editor Report · Decision Letter 1]

5 Jul 2023

JAK-inhibitors and risk on serious viral infection, venous thromboembolism and cardiac events in patients with Rheumatoid Arthritis: A protocol for a prevalent new-user cohort study using the Danish nationwide DANBIO register.

PONE-D-23-07128R1

Dear Dr. Burden,

We’re pleased to inform you that your manuscript has been judged scientifically suitable for publication and will be formally accepted for publication once it meets all outstanding technical requirements.

Kind regards,

Ryu Watanabe, M.D., Ph.D.

Academic Editor

PLOS ONE

Additional Editor Comments (optional):

The authors responded adequately to the comments and suggestions raised by the reviewers. I think the manuscript is now acceptable to this journal.

---

## [Editor Report · Acceptance letter]

19 Jul 2023

PONE-D-23-07128R1 

JAK-inhibitors and risk on serious viral infection, venous thromboembolism and cardiac events in patients with rheumatoid arthritis: A protocol for a prevalent new-user cohort study using the Danish nationwide DANBIO register 

Dear Dr. Burden:

I'm pleased to inform you that your manuscript has been deemed suitable for publication in PLOS ONE. Congratulations! Your manuscript is now with our production department. 

Kind regards, 

on behalf of

Dr. Ryu Watanabe 

Academic Editor

PLOS ONE